# Effects of Different Opioid Drugs on Oxidative Status and Proteasome Activity in SH-SY5Y Cells

**DOI:** 10.3390/molecules27238321

**Published:** 2022-11-29

**Authors:** Laura Rullo, Francesca Felicia Caputi, Loredana Maria Losapio, Camilla Morosini, Luca Posa, Donatella Canistro, Fabio Vivarelli, Patrizia Romualdi, Sanzio Candeletti

**Affiliations:** 1Department of Pharmacy and Biotechnology, Alma Mater Studiorum—University of Bologna, Via Irnerio 48, 40126 Bologna, Italy; 2Department of Pharmacology and Experimental Therapeutics, Boston University, 700 Albany Street, Boston, MA 02118, USA

**Keywords:** ROS, SOD, proteasome, morphine, buprenorphine, tapentadol

## Abstract

Opioids are the most effective drugs used for the management of moderate to severe pain; however, their chronic use is often associated with numerous adverse effects. Some results indicate the involvement of oxidative stress as well as of proteasome function in the development of some opioid-related side effects including analgesic tolerance, opioid-induced hyperalgesia (OIH) and dependence. Based on the evidence, this study investigated the impact of morphine, buprenorphine or tapentadol on intracellular reactive oxygen species levels (ROS), superoxide dismutase activity/gene expression, as well as β2 and β5 subunit proteasome activity/biosynthesis in SH-SY5Y cells. Results showed that tested opioids differently altered ROS production and SOD activity/biosynthesis. Indeed, the increase in ROS production and the reduction in SOD function elicited by morphine were not shared by the other opioids. Moreover, tested drugs produced distinct changes in β2(trypsin-like) and β5(chymotrypsin-like) proteasome activity and biosynthesis. In fact, while prolonged morphine exposure significantly increased the proteolytic activity of both subunits and β5 mRNA levels, buprenorphine and tapentadol either reduced or did not alter these parameters. These results, showing different actions of the selected opioid drugs on the investigated parameters, suggest that a low µ receptor intrinsic efficacy could be related to a smaller oxidative stress and proteasome activation and could be useful to shed more light on the role of the investigated cellular processes in the occurrence of these opioid drug side effects.

## 1. Introduction

Chronic pain represents one of the major health issues in our society. Although mortality rates are highest for other pathologies, this condition is among the main sources of human suffering and disability that profoundly impacts patients’ quality of life [1]. Despite research advancement and the suggestions of new druggable targets for acute and chronic pain treatment, opioids still represent gold-standard analgesics. However, their prolonged use is often hampered by several adverse side effects including the development of analgesic tolerance, opioid-induced hyperalgesia (OIH) and dependence [2]. Although these phenomena are not yet completely understood, molecular alterations in opioid receptor signaling, neurotransmitter release changes, as well as glia and microglia activation have been suggested as possible mechanisms involved in the development of side effects related to chronic opioid treatment [3,4,5,6,7,8,9,10]. Furthermore, a role of oxidative stress in the induction of morphine adverse events has been suggested [11,12,13]. In this regard, substantial data showed that the production of reactive species could participate in chronic pain establishment as well as in the development of opioid drugs side effects, probably through a neuroinflammation process [14,15,16]. Indeed, even though low levels of reactive oxygen species (ROS) are fundamental for the activation of some cellular pathways [17], increased ROS levels can induce biomolecular damage and also trigger the activation of transcription factors involved in the control of many inflammatory genes and proteins [18]. In this context, the degradation of non-functional proteins represents a fundamental cellular process necessary to protect cells from oxidative damage and to maintain redox balance. Although some evidence suggests that an oxidative stress response could induce chaperone-mediated autophagy [19], proteasome represents the major proteolytic machinery involved in the degradation of the oxidized and misfolded proteins [18,20,21,22]. The 20S proteasome is a barrel-shaped complex made up through the assembly of two outer α-rings and two inner β-rings. The α-rings are both characterized by seven α regulatory protein (α_1–7_), while the β-rings consist of seven different β subunits (β_1–7_) including the β1, β2, and β5 subunits known to possess caspase, trypsin, and chymotrypsin-like activities, respectively. As suggested by numerous reports, the 20S proteasome is mainly responsible for the degradation of non-ubiquitinated, misfolded and oxidized proteins [18,22]. However, the 20S proteasome can be capped by one or two terminal 19S regulatory particle(s) to form the 26S or 30S proteasome [18,23]. In contrast to the 20S proteasome, the latter are mainly implicated in ubiquitin/ATP-dependent protein degradation [23,24].

Given the pivotal role of this multi-catalytic complex in the maintenance of cellular proteostasis, evidence highlighted its involvement in the pathogenesis of several human diseases [25,26]. The role of proteasome activation in morphine-induced tolerance and dependence has been suggested in particular in recent decades [27,28,29,30]. Indeed, the ability of proteasome inhibitors to prevent/revert both tolerance and opioid-induced hyperalgesia has been shown [29,31], thus supporting the possible implication of this degradation complex in the development of the above-mentioned phenomena. On this basis, we aimed to investigate the effects of selected opioid ligands in biological processes associated with oxidative stress and proteasome activation. To this end, the ability of morphine, buprenorphine and tapentadol to affect intracellular reactive oxygen species (ROS) levels, superoxide dismutase activity/gene expression, as well as modulate β2 and β5 subunit proteasome activity/biosynthesis was assessed in SH-SY5Y cell cultures.

## 2. Results

### 2.1. Intracellular ROS Levels

Opioid ligand ability to affect intracellular ROS production was assessed in SH-SY5H cells treated for 2, 5, 24 or 48 h with 10 µM morphine, 0.25 µM buprenorphine or 10 µM tapentadol, by the DCFH-DA assay. No significant differences between untreated control and morphine-, buprenorphine- or tapentadol-treated cells were detected at 2 and 5 h (*p* > 0.05 vs. control) (Figure 1a,b). Interestingly, a statistically significant increase in ROS levels was observed at later intervals for morphine-treated cell cultures only (24 h: *p* < 0.05 and 48 h: *p* < 0.01 vs. control) (Figure 1c,d).

Based on these results and to assess the involvement of μ opioid receptor (MOR) activation in morphine-mediated ROS generation, cells were co-treated with 10 μM of the non-selective opioids antagonist, naloxone, or with 10 μM of the selective MOR antagonist, β-funaltrexamine (β-FNA). The analysis at 24 h, confirmed the ability of morphine to induce an increase in ROS intracellular generation (* *p* < 0.05 vs. control) (Figure 1 insert). Interestingly, results showed that morphine effects on ROS generation were prevented by both non-selective and selective MOR antagonist. Indeed, no significant changes from control values were observed in cell cultures co-exposed to naloxone and morphine (*p* > 0.05 vs. control) or to β-FNA and morphine (*p* > 0.05 vs. control) (Figure 1 insert).

According to these data, showing alteration in cell oxidative status only after 24 and 48 h of drug exposure, subsequent analyses of different drug effects upon SOD and proteasome activity/gene expression were performed at these longer intervals.

### 2.2. SOD Activity and Gene Expression

Since an increase in ROS is often associated with a dysfunction of SOD antioxidant enzyme activity [32], we evaluated whether morphine, buprenorphine or tapentadol could affect these parameters at the selected time points. Data showed a statistically significant decrease in SOD enzymatic activity in morphine-treated cells at 24 h (*p* < 0.05 vs. controls) (Figure 2a), while no significant changes in this enzymatic activity were found after 48 h of treatment (*p* > 0.05 vs. control) (Figure 2b). No significant changes in this parameter were detected after other drug treatment, at both intervals (*p* > 0.05 vs. control) (Figure 2a,b).

Similarly to as observed for SOD enzymatic activity, changes in SOD1 biosynthesis were assessed at the 24 h observation interval only. In particular, a significant up-regulation of SOD1 mRNA levels was observed in SH-SY5H cells treated for 24 h both with morphine or buprenorphine (*p* < 0.0001 vs. control) (Figure 3a).

### 2.3. Proteasome Activity and β2/β5 Subunit Biosynthesis

Given relationships between cell oxidative stress and proteasome function/regulation [18], proteasomal enzymatic activity as well as specific subunit de novo synthesis were evaluated in our experimental conditions. Results indicated that the investigated opioid drugs differently alter β2 trypsin-like and β5 chymotrypsin-like proteasome activities at the two observation intervals.

Indeed, morphine treatment was able to induce a significant increase in β5 proteolytic activity at 24 h (β5: *p* < 0.01 vs. control) (Figure 4a,c) and an enhancement of both β2 and β5 proteasome activity at 48 h (β2: *p* < 0.01 vs. control; β5: *p* < 0.0001 vs. control), in SH-SY5Y cells (Figure 4b,d). Differently, a reduction in β5 enzymatic activity was observed after 24 h of buprenorphine treatment (*p* < 0.001 vs. control) and a decrease in β2 activity was detected after 48 h exposure to the same drug (*p* < 0.0001 vs. control) (Figure 4b,c). Tapentadol did not significantly change the enzymatic activity of both β subunits at any assessment intervals (*p* > 0.05 vs. control) (Figure 4a–d).

Gene expression results showed that the investigated drugs also affect β2 and β5 subunit biosynthesis. In particular, morphine induced a significant down-regulation of β2 subunit gene expression both at 24 and 48 h (*p* < 0.01; *p* < 0.001 vs. control) (Figure 5a,b), while it caused an increase in β5 mRNA levels after 48 h (*p* < 0.05 vs. control) (Figure 5d). As regards the impact of buprenorphine and tapentadol on β subunits gene expression, data showed that these molecules decreased β2 and β5 mRNA levels after 24 h (*p* < 0.01; *p* < 0.001 vs. control) (Figure 5a,c) and that β5 subunit down-regulation was maintained at 48 h only after tapentadol treatment (*p* < 0.01 vs. control) (Figure 5d).

## 3. Discussion

Several mechanisms have been suggested in the development of the most important opioid-related side effects [7,8,9,10]. Among the plethora of molecular mechanisms proposed, attention has been also focused on oxidative stress and proteasome function [12,13,14,15,31].

Indeed, the regulation of cellular ROS is important since low levels of these reactive species are involved in the modulation cell proliferation, differentiation and act as signaling messengers. However, their excessive production causes negative cell effects including the dysregulation of cell macromolecule homeostasis with the subsequent involvement of proteasome that plays a crucial role in the maintenance of redox balance by recognizing and removing the oxidatively modified or damaged proteins [18,22].

In order to explore opioids’ relationship with oxidative stress and proteasome regulation, the aims of this study were to investigate the effects of morphine, buprenorphine and tapentadol on ROS generation, superoxide dismutase activity/gene expression, as well as on β2 and β5 subunit proteasome activity/biosynthesis in human SH-SY5Y cell line.

Data here reported showed that the selected drugs differently alter ROS production level. Indeed, the ROS-increasing effect of morphine is not shared by the other opioid analgesics here investigated, thus suggesting that the specific drug pharmacological profile can influence this parameter. In accordance with previous studies that indicated morphine’s ability to induce an oxidative stress in SH-SY5Y cells [33,34], our findings support the hypothesis that agonist stimulation of MOR could contribute to the increase in intracellular ROS. Indeed, the MOR-agonist-induced reactive oxygen species generation is completely blocked by the non-selective MOR antagonist naloxone and the selective MOR antagonist β-funaltrexamine (β-FNA). Although ROS can act as signaling molecules [35] evidence has shown their involvement in the promotion of desensitization, down-regulation and tolerance for GPCRs, including MOR [33,36,37,38,39], thus supporting their pivotal role in morphine side effects.

Together with alteration in ROS generation, a significant decrease in SOD enzymatic activity and an increase in SOD1 mRNA levels were observed in SH-SY5Y cells after 24 h of morphine exposure. It is conceivable that the increase in the SOD1 gene expression might represent a cellular adaptive mechanism aimed to raise enzyme availability and to counteract ROS production and the subsequent oxidative damage. In line with our finding, an increase in this enzyme isoform levels has been observed in the nucleus accumbens of non-human primates subjected to a prolonged morphine treatment [40]. However, morphine effects on cell antioxidant machinery are still controversial since both an increase or a decrease in SOD activity or biosynthesis have been reported in different cell lines and at different concentrations [12,41]. Taken together, these results suggest that morphine could be able to produce a higher ROS-mediated neurotoxic effect than the other investigated compounds [42]. In this context, it is interesting to note that oxidative processes could be related to the development of analgesic tolerance and probably dependence since some studies showed that the administration of molecules acting as antioxidant or SOD-mimetics prevent or counteract morphine tolerance in different animal models [43,44,45,46].

Our data also indicated that morphine, buprenorphine, and tapentadol produced different changes in β2 trypsin-like and β5 chymotrypsin-like activities. Overall data analysis, in accordance with previous studies, showed that morphine was able to increase the proteolytic activity after prolonged exposure [47]. Instead, a different picture was observed for buprenorphine and tapentadol, with buprenorphine reducing proteasome activity after prolonged exposure intervals and tapentadol unable to induce significant alterations at any assessment time point.

Given the strong correlation between proteasome and oxidative stress [18,21,22], the increase in proteasome activity could be related to the rise of oxidized proteins. In fact, it is known that an excessive ROS production and the subsequent protein oxidation induces a rapid activation of the proteasome degradation complex and the de novo synthesis of its proteolytic subunits [21,22]. In this regard, gene expression data overall showed that tested compound are able to differently affect β subunits biosynthesis. Interestingly, morphine promoted an increase, whereas buprenorphine and tapentadol caused a reduction in mRNA levels for β5 subunit which represents the more active proteasome subunit and the more clearly induced during oxidative stress processes [48]. However, given the long half-life of this complex, additional studies will be useful to better clarify if the gene expression changes in proteasome subunits here observed are related to their bio-availability.

Different preclinical and clinical studies pointed out the possibility to identify two distinct clusters of opioid drugs, based on their intrinsic efficacy and side effect profile (e.g., tolerability and abuse liability). On this basis, it is possible that buprenorphine and tapentadol, showing lower µ receptor intrinsic efficacy than morphine [49,50], might induce less oxidative stress and proteasome activation possibly associated with a more favorable side effect profile.

However, since proteasome is involved either in oxidized and polyubiquitinated protein degradation, it is conceivable that the enhancement of 20S proteasome activity after morphine treatment might be also related to an increase in polyubiquitinated proteins. In this regard, previous studies suggested a crucial role for the ubiquitin/proteasome pathway in agonist-induced down-regulation and in basal turnover of μ-opioid receptors [51]. In addition, Moulédous and coworkers showed that the sustained MOR activation by morphine promotes the proteasomal degradation of Gβγ which, in turn, contributes to adenylate cyclase sensitization, a hallmark of opiate dependence [52]. In addition to in vitro results, animal studies also highlighted the involvement of ubiquitin/proteasome system in the development of morphine tolerance and in addictive behavior [30,31]. Indeed, it has been demonstrated that proteasome inhibitors prevent the development of morphine tolerance inhibiting the proteasomal degradation of spinal glutamate transporter and the decrease in spinal glutamate uptake activity [31]. Moreover, the intra-nucleus accumbens infusion of proteasome inhibitors seems to prevent morphine preference in mice probably preventing the proteasomal degradation of polyubiquitinated proteins in this specific area [30].

These results give further insights into the relationships between opioid drugs and oxidative stress. Even though some data suggested a neuroprotective role of specific concentrations of morphine [53,54,55], our data support the view of a pro-oxidative effect [12,13,33,56], of this analgesic alkaloid. Data here presented provide a new comparison of different opioid drug ability to affect cell oxidative status and proteasome activity/biosynthesis and highlight the higher impact of morphine on these parameters.

In light of literature data about the effect of antioxidants and proteasome inhibitors upon morphine side effects [31,43,57,58,59,60], our data could be useful to better understand the involvement of the investigated cell processes in the different opioid drugs’ adverse effects.

Since a role of peroxynitrite, the product of the interaction between superoxide and nitric oxide, in morphine antinociceptive tolerance has been proposed [15], future studies will be aimed at investigating the relative ability of tested ligands to induce the production of this highly reactive species.

## 4. Materials and Methods

### 4.1. Cell Culture and Treatments

Human SH-SY5Y neuroblastoma cells purchased from ICLC-IST (Genoa, Italy) were cultured in Dulbecco’s modified Eagle medium (DMEM), supplemented with 10% (*v*/*v*) fetal bovine serum (FBS), 100 units/mL penicillin, 100 μg/mL streptomycin and 2 mM glutamine. Cells were incubated at 37 °C in a humidified atmosphere containing 5% CO2 and were allowed to reach 80% confluence before starting any treatment. All reagents employed for cell culture were purchased from Lonza (Milan, Italy). SH-SY5Y cells were exposed to 10 μM morphine (Carlo Erba, Cornaredo, Italy), 0.25 μM buprenorphine (Sigma-Aldrich, Milan, Italy) or 10 μM tapentadol (Grünenthal GmbH, Aachen, Germany) in accordance with previous studies regarding morphine and tapentadol [37,61,62]. As regards buprenorphine, we referred to the drug potency of buprenorphine vs. morphine (25–100% more potent than morphine); therefore, we decided to use 0.25 μM [63].

Moreover, to test the influence of MOR activation on ROS generation, cells were pretreated with 10 μM naloxone (non-selective opioid antagonist, Tocris Biotechne, Milan, Italy) or with 10 μM β-funaltrexamine, β-FNA (MOR selective antagonist, Sigma-Aldrich, Milan, Italy), 30 min before morphine. Four/six biological replicates per treatment were utilized in each experiment.

### 4.2. Intracellular Reactive Oxygen Species Production

Intracellular reactive oxygen species were originally measured with the 2,7-dichlorofluorescin diacetate (DCFH-DA) assay, OxiSelect™ Intracellular ROS Assay Kit (Cell Biolabs, San Diego, CA, USA). The assay was carried out in accordance with the manufacturer’s instruction. Briefly, SH-SY5Y cells were cultured in black 96-well plates. 10 μM DCFH-DA was then dissolved in medium containing 1% FBS and 100 µL of this solution was added to each well. Cells were incubated for 60 min in order to allow cellular incorporation. Thereafter, the original medium was discarded, and 10 µM morphine, 0.25 µM buprenorphine or 10 µM tapentadol was added to the cell medium, culturing for 2, 5, 24, or 48 h. Then, DCF fluorescence intensity (RFU) was read at 37 °C in a fluorescence plate reader (GENios Tecan, Männedorf, Switzerland) with an emission wavelength of 530 nm and an excitation wavelength of 480 nm. Results are expressed as the percentage of RFU relative to controls. The determination of ROS generation in morphine-treated cells pretreated with MOR non-selective or selective antagonists was assessed at 24 h by electronic paramagnetic resonance (EPR) as previously described [64]. After the treatment, cells were collected and centrifuged at 300× *g* for 5 min. Pellets were suspended in 1 mL of standard physiological solution containing the hydroxylamine “spin probe” (bis(1-hydroxy-2,2,6,6-tetramethyl-4-piperidinyl) decandioate dihydrochloride) (1 mM), incubated at 37 °C for 5 min and then frozen at −80 °C until EPR analysis.

### 4.3. Protein Extraction

After treatments (24 or 48 h), cells were homogenized in lysis buffer (150 mM NaCl, 50 mM HEPES pH 7.5, 5 mM EDTA, 2 mM ATP, 1% Triton; Sigma-Aldrich, Milan, Italy) and centrifuged at 14,000× *g* at 4 °C for 15 min, as previously described [65]. Protein concentration was determined by using the Pierce BCA protein assay kit (Thermo Fischer Scientific, Waltham, MA, USA) and supernatants were aliquoted and kept at −80 °C until SOD and proteasome activities assays.

### 4.4. SOD Activity Assay

SOD activity was determined using the SOD assay kit (Sigma-Aldrich, Milan, Italy). Briefly, the assay is based on the use of the highly water-soluble tetrazolium salt, WST-1 [2-(4-Iodophenyl)-3-(4-nitrophenyl)-5-(2,4-disulfophenyl)-2*H*-tetrazolium monosodium salt], which forms a water-soluble formazan dye upon reduction with a superoxide anion, with a reduction rate that is inversely proportional to SOD activity. In accordance with the manufacturer’s instructions and previous studies [66], 25 μg of proteins were assayed in triplicate for each sample. After incubation at 37 °C for 20 min, absorbance was measured at 450 nm using a plate reader fluorometer (GENios Tecan, Männedorf, Switzerland).

### 4.5. Proteasome Activity Assay

Proteasome β2 trypsin-like and β5 chymotrypsin-like activities were analyzed by monitoring the cleavage of benzyloxycarbonyl-Ala-Arg-Arg-7-amino-4-methylcoumarin (Z-ARR-AMC) and succinyl-Leu-Leu-Val-Tyr-7-amino-4-methylcoumarin (Suc-LLVY-AMC) (both purchased from Merck Millipore, Roma, Italy), respectively, using 25 μg of cell lysate proteins per sample [67]. The assay is based on the detection of the fluorophore 7-amino-4-methylcoumarin (AMC) after cleavage from the labeled substrates. The free AMC fluorescence was quantified at 380 nm excitation and 460 nm emission wavelengths using a plate reader fluorometer (GENios Tecan, Männedorf, Switzerland). Data are expressed as the percentage of RFU relative to controls.

### 4.6. RNA Isolation and qRT-PCR

After treatments (24 or 48 h), total RNA was isolated using the TRIZOL reagent (Life Technologies, Carlsbad, CA, USA) according to the method of Chomczynski and Sacchi [68] and its integrity was checked by 1% agarose gel electrophoresis. In brief, the amounts of RNA were determined by measuring optical densities and only RNA samples with an OD260/OD280 1.8 < ratio < 2 were used. Total RNA was reverse transcribed as previously described [69,70]. Quantitative real-time PCR was performed on a StepOne Real-Time PCR System (Life Technologies, Carlsbad, CA, USA) using TaqMan Gene Expression Master Mix (ThermoFisher Scientific, Waltham, MA, USA), to analyze SOD1 (Hs 00533490_m1, FAM), β2 (Hs 01002946_m1, FAM), β5 (Hs 00605652_m1, FAM). All samples were run in triplicate and were normalized to the endogenous reference gene glyceraldehyde—3-phosphate dehydrogenase (GAPDH) (Hs 03929097_g1, VIC).

TaqMan Probe sequences were purchased from ThermoFisher Scientific, Waltham, MA, USA.

### 4.7. Statistical Analysis

Biochemical data have been initially evaluated by Shapiro-Wilk tests to confirm the normality of the distribution and by Grubb’s test to identify outliers, then they were analyzed by one-way ANOVA followed by Dunnett’s multiple comparison as a post hoc test. Results were expressed as the mean ± SEM of three/six biological replicates per treatment. Statistical analysis was performed using the GraphPad Prism software package (v9 for Windows, GraphPad Software, San Diego, CA, USA). The threshold for statistical significance was set at *p* < 0.05.

## 5. Conclusions

In summary, the results highlighted the different ability of opioids to alter redox balance and proteasome function in SH-SY5Y cells. Even though additional studies are still necessary to elucidate the involvement of these processes in opioid action, our results offer new information on different opioid drug effects on oxidative stress and proteasome and support the view that the modulation of the redox balance might be a promising pharmacological approach for chronic pain treatment.

## Figures and Tables

**Figure 1 molecules-27-08321-f001:**
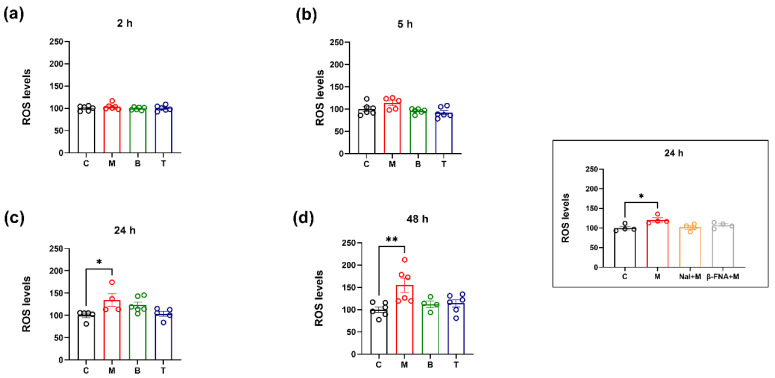
ROS production after treatment with 10 µM morphine (M), 0.25 µM buprenorphine (B) or 10 µM tapentadol (T) in SH-SY5Y cells at different time points (**a**–**d**). The right framed insert reports ROS production after cotreatment with 10 µM naloxone (Nal) + 10 µM morphine (M) or 10 µM β-funaltrexamine (β-FNA) + 10 µM morphine (M) in SH-SY5Y cells at 24 h. Data, reported as the mean ± SEM of four/six biological replicates for each treatment, are expressed as the percentage of relative fluorescence (**a**–**d**) or the percentage of the intensity of the first spectra line (insert) compared to control (C) and are analyzed by one-way ANOVA followed by Dunnett’s multiple comparisons test (* *p* < 0.05; ** *p* < 0.01 vs. control).

**Figure 2 molecules-27-08321-f002:**
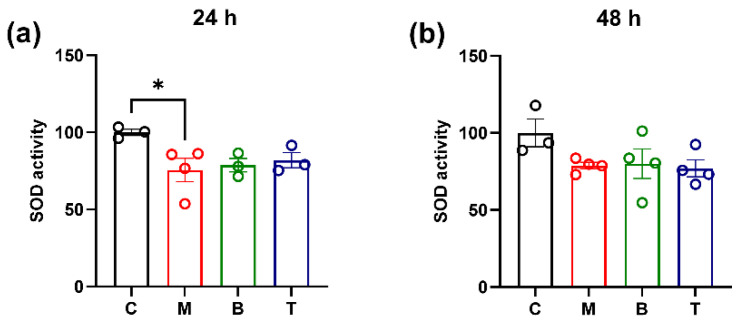
SOD activity after treatment with 10 µM morphine (M), 0.25 µM buprenorphine (B) or 10 µM tapentadol (T) in SH-SY5Y cells at 24 h (**a**) and 48 h (**b**). Data, reported as the mean ± SEM of three/four biological replicates for each treatment, are expressed as the percentage of inhibition rate (absorbance) compared to control (C) and analyzed by one-way ANOVA followed by Dunnett’s multiple comparisons test (* *p* < 0.05 vs. control).

**Figure 3 molecules-27-08321-f003:**
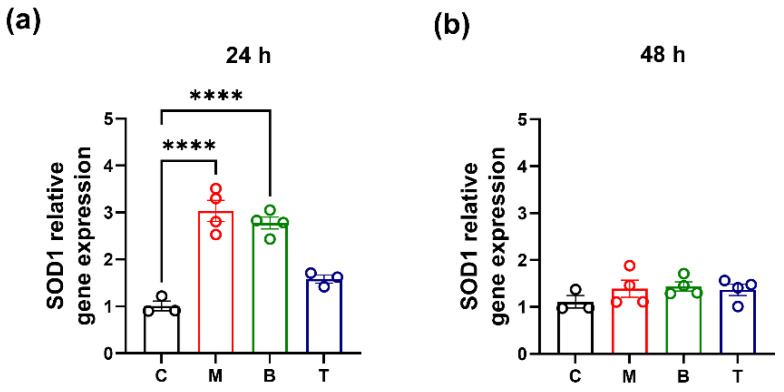
SOD1 relative gene expression after treatment with 10 µM morphine (M), 0.25 µM buprenorphine (B) or 10 µM tapentadol (T) in SH-SY5Y cells at 24 h (**a**) and 48 h (**b**). Data, expressed as the mean ± SEM of three/four biological replicates for each treatment, represent 2^−DDCt^ values and are analyzed by one-way ANOVA followed by Dunnett’s multiple comparisons test (**** *p* < 0.0001 vs. control (C)).

**Figure 4 molecules-27-08321-f004:**
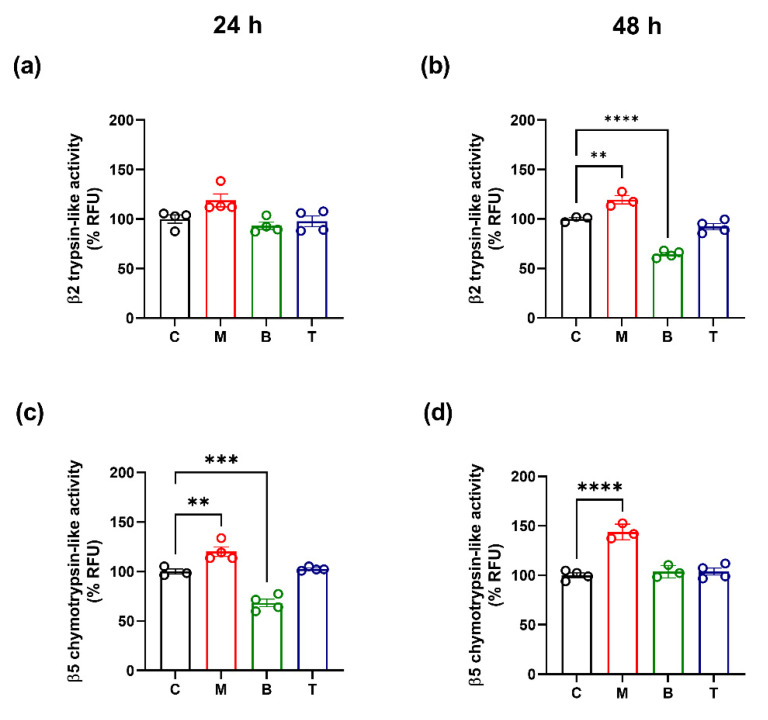
β2 trypsin-like (**a**,**b**) and β5 chemotrypsin-like (**c**,**d**) activities after treatment with 10 µM morphine (M), 0.25 µM buprenorphine (B) or 10 µM tapentadol (T) in SH-SY5Y cells at 24 h (**a**,**c**) and 48 h (**b**,**d**). Data, reported as the mean ± SEM of three/four biological replicates for each treatment, are expressed as the percentage of relative fluorescence (RFU) compared to control (C) and analyzed by one-way ANOVA followed by Dunnett’s multiple comparisons test (** *p* < 0.01; *** *p* < 0.001 vs. control; **** *p* < 0.0001 vs. control).

**Figure 5 molecules-27-08321-f005:**
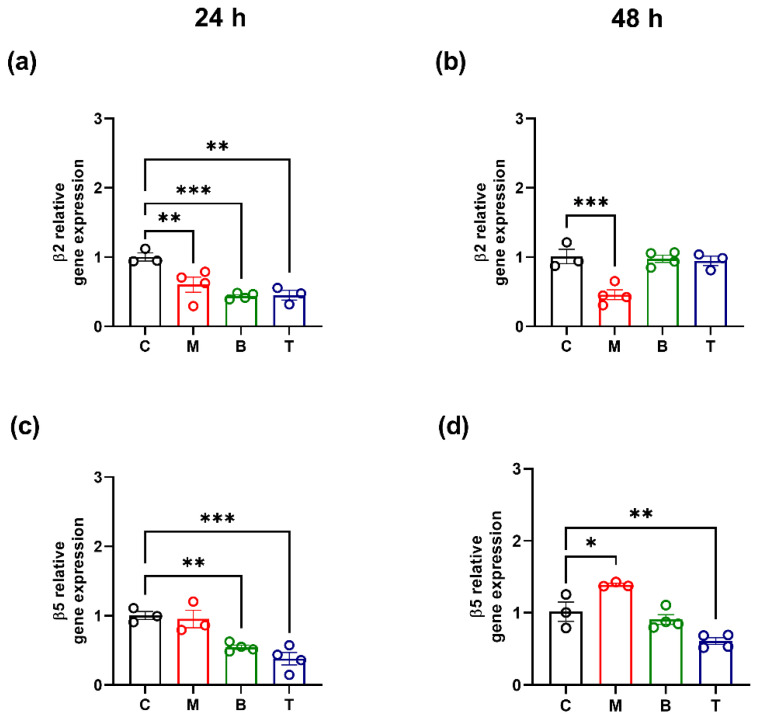
Relative gene expression of β2 (**a**,**b**) and β5 (**c**,**d**) subunits after treatment with 10 µM morphine (M), 0.25 µM buprenorphine (B) or 10 µM tapentadol (T) in SH-SY5Y cells at 24 h (**a**,**c**) and 48 h (**b**,**d**). Data, expressed as the mean ± SEM of three/four biological replicates per treatment, represent 2^−DDCt^ values and are analyzed by one-way ANOVA followed by Dunnett’s multiple comparisons test (* *p* < 0.05; ** *p* < 0.01; *** *p* < 0.001 vs. control (C)).

## Data Availability

Data supporting the finding of this study are available upon reasonable request.

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
