# Peer review of "Effects of Different Opioid Drugs on Oxidative Status and Proteasome Activity in SH-SY5Y Cells"

_molecules, 2022, doi:10.3390/molecules27238321_

Round 1

Reviewer 1 Report

In the paper entitle “Effects of different opioid drugs on oxidative status and proteasome regulation in SH-SY5Y cells”, Rullo and co-authors draw a comparison of the effect of different opioid drugs on redox unbalance and proteasome dysregulation.

The paper is well written and presented, however it suffers from some criticisms concerning the proteasome analysis that not support the conclusions raised by the authors and need to be addressed before further considering for publication.

Minor points:

1)      For readers less expert in the field, it would be relevant to clarify why different concentrations of the drugs were actually used in vitro;

2)      Some introduction about proteasome would be useful for the readership;

Major points:

The profile of proteasome subunit expression shows some inconsistencies with activity in vitro.

My personal comment, which I hope can help the authors revising the manuscript, is that to measure subunits expression is not enough to draw conclusion about the effective bio-availability of subunits and, more importantly, the effective incorporation of subunits into fully assembled 20S particles.

Furthermore, it must be considered that proteasome has a very long half-life. Hence, downregulation of some subunits may have no significant effects on proteasome activity, as it actually looks to happen in the case presented (at least for beta2 subunit).

Therefore, to better address the possible dysregulation of proteasome the following experiments are required:

-          Western blotting (semiquantitative analysis) of subunits indicated and additional subunits (I would suggest one out of the seven alpha subunits and one or two 19S subunits); This analysis must include also the determination of poly-ubiquitinated proteins. This is commented in the discussion, but is relevant for data interpretation.

-          Unlike LLVY-amc, which shows a good specificity for chymotrypsin-like activity, fluorogenic substrates of trypsin-like activity show a high rate of non-specific cleavage by intracellular peptidases. To more accurately determine the trypsin-like activity, to compare data in the presence/absence of a proteasome inhibitor is mandatory. Commercial inhibitors have lower efficacy for trypsin-like sites but they work well at high concentration;

-        A native gel electrophoresis assay (optional but strongly suggested) to visualize the proteasome particles. I am concerned about the possibility that different pathways come into play (autophagy?) and overall interpretation of data is affected by them.

Author Response

Reviewer 1

Comments and Suggestions for Authors

In the paper entitled “Effects of different opioid drugs on oxidative status and proteasome regulation in SH-SY5Y cells”, Rullo and co-authors draw a comparison of the effect of different opioid drugs on redox unbalance and proteasome dysregulation.

The paper is well written and presented, however it suffers from some criticisms concerning the proteasome analysis that not support the conclusions raised by the authors and need to be addressed before further considering for publication.

Minor points:

  • For readers less expert in the field, it would be relevant to clarify why different concentrations of the drugs were actually used in vitro;

  1. The reason why we utilized different concentrations of the drugs used in this study in vitro has been reported in Material and Methods section (paragraph 4.1). Namely, we clarified that we used morphine concentration at 10 μM because it has been used many times in several publications from our and other laboratories (see refs cited in the manuscript). The same happened for tapentadol. As regard buprenorphine we referred to the drug potency of buprenorphine vs morphine ( 25-100 times more potent than morphine), therefore we decided to use 0.25 μM ( ref PMID: 26672499)

  • Some introduction about proteasome would be useful for the readership;

A         We thank the Reviewer for the suggestion and we agree that it is useful for the readers. Accordingly, we added some information about proteasome structure and activity in the Introduction section (see: page 2, lines 54-72.)

Major points:

The profile of proteasome subunit expression shows some inconsistencies with activity in vitro.

My personal comment, which I hope can help the authors revising the manuscript, is that to measure subunits expression is not enough to draw conclusion about the effective bio-availability of subunits and, more importantly, the effective incorporation of subunits into fully assembled 20S particles.

A    We thank the reviewer for the comments and we really appreciate his/her useful suggestion. We perfectly understand the perplexity about the gene expression analysis of β subunits and their effective bio-availability and incorporation into the fully 20S particles.

 According with his/her observations we added this limitation in the discussion (page 11, lines 270-273). Taking into account the exploratory nature of this study, the title has been revised. Indeed, the majority of our observations about the involvement of proteasome in morphine-mediated effects are mainly referred to proteasome activity rather than to its regulation. Furthermore, it must be considered that proteasome has a very long half-life. Hence, downregulation of some subunits may have no significant effects on proteasome activity, as it actually seems to happen at least for beta2 subunit.

We think that measuring gene expression for proteasome subunits may help anyway to understand what it is happening after the exposure to the opiate drugs vs proteasome activity.

Therefore, to better address the possible dysregulation of proteasome the following experiments are required:

-     Western blotting (semiquantitative analysis) of subunits indicated and additional subunits (I would suggest one out of the seven alpha subunits and one or two 19S subunits); This analysis must include also the determination of poly-ubiquitinated proteins. This is commented in the discussion, but is relevant for data interpretation.

A    In our opinion, all the suggested experiments represent a very interesting aspect of the study on proteasome that is however another aspect of drug design that we decided to run in our study. I am convinced that our findings are useful information that can show to the scientific community that some changes occur after exposure to different opiate drugs. These can be useful both for us for going on in our study but also for other groups to design a new study that in any case takes months.

However, there are several evidence which showed the crucial role played by the ubiquitin/proteasome pathway in opioid agonist effects.  Indeed, in vitro studies already suggested that the protein polyubiquitination and the consequent proteosomal degradation are strongly involved in the down-regulation and basal turnover of μ-opioid receptors as well as in heterodimeric G protein, regulator G protein signaling protein 4 (RGS4)  and glutamate transporters degradation (PMID: 11152677; PMID: 15901846; PMID: 21209077; PMID: 18539596; PMID: 18986766). In addition, it has been also showed the intra-nucleus accumbens infusion of proteasome inhibitors seems to prevent morphine preference in mice, probably preventing the proteasomal degradation of polyubiquitinated proteins in this specific area (PMID: 23169349).

Considering these evidence showing the involvement of ubiquitin-proteasome system in some morphine effects, we expanded this section in the discussion( pag 11, lines 274-293) to emphasize that the increase of 20s proteasome activity observed in our study could be also related to an increase of polyubiquitinaded proteins.

-     Unlike LLVY-amc, which shows a good specificity for chymotrypsin-like activity, fluorogenic substrates of trypsin-like activity show a high rate of non-specific cleavage by intracellular peptidases. To more accurately determine the trypsin-like activity, to compare data in the presence/absence of a proteasome inhibitor is mandatory. Commercial inhibitors have lower efficacy for trypsin-like sites but they work well at high concentration.

A    We thank the Reviewer for the comment. We understand Reviewer’s concern about the trypsin like activity and certainly we will consider this aspect in future studies. However, according with a previous paper (PMID: 156617369) showing the major importance of β5 in response to oxidative stress and taking into account that especially after morphine treatment both the subunits activity appear to be upregulated, we think that the overall analysis gives useful information about the ability of different opioids to affect the proteasome activity.

-        A native gel electrophoresis assay (optional but strongly suggested) to visualize the proteasome particles. I am concerned about the possibility that different pathways come into play (autophagy?) and overall interpretation of data is affected by them.

A          We understand the reviewer’ s concerns. Indeed, some studies have been already suggested the involvement of lysosomal degradation pathway in the opioid receptor trafficking, especially for delta-receptor (PMID: 10753919; PMID: 10366739) However, it has been also showed by Chaturvedi and coworkers that that the opioid agonist-induced down-regulation of opioid receptors can be blocked by proteasome inhibitors but not by lysosomal, calpain, or caspase protease inhibitors. In addition, Proteasome inhibitors, but not lysosomal protease inhibitors, seem to be able to increase steady-state levels of μ and δ opioid receptors (PMID: 11152677).

Therefore, as suggest by these authors it is possible that ubiquitin/proteasome pathway operates upstream of trafficking to lysosomes to initiate receptor down-regulation (PMID: 11152677).

Besides to this observation, also morphine-induced post-transcriptional down-regulation of the glutamate transporter EAAC1 (PMID: 18539596) was blocked by the selective proteasome inhibitor MG-132 or lactacystin but not the lysosomal inhibitor chloroquine, thus suggesting the major involvement of proteasome system in morphine-mediated effects. However, taking into account the importance of lysosomal degradation in cellular trafficking, we will certainly investigate the possible involvement of other pathways in opioid-mediated effects.

Lastly, we would like to tell the Reviewer 1 that we could do only some experiments requested by Reviewer 2, due to the time available for the revision- (initially 10 days, plus 15 days).

We really hope that the Reviewer can understand that all the required new experiments would have employed more than 3-4 months, considering also the time spending in growing cells and that we would continue our work on this project and leave all the interesting experiments suggested for a new manuscript on next year.

Reviewer 2 Report

In this manuscript, Rullo et al. examined the effect of opioid agonist exposure on  ROS levels, SOD activity/gene expression and proteosome activity in SH-SY5Y cells. The authors found that ROS levels increased after 24 and 48 hr morphine exposure. SOD activity also decreased after 24 hr and not 48. SOD1 gene expression increased significantly at 24 hr, but not 48 hr. Finally, beta2 trypsin-like and b5 chemotrypsin-like activity were increased with morphine but decreased with buprenorphine.

There are some significant issues with this paper. 

1) The authors used a really high morphine and tapentadol concentration. At these levels it is likely that the drugs had nonspecific effects. Further, the time of exposure is rather long because opioid receptor internalization very likely occurred before the 24 hr period expired. 

2) To show convincing evidence that the actions were mediated via mu opioid receptors, the authors have to show that they can pharmacologically block the effect of the opioids.

3) The cell type employed has other opioid receptors. How do the authors rule out their involvement when using such high opioid concentrations?

4) Authors must show scatter plots for all their data. Bar graphs are not convincing.

5) The authors should also change their "statistical analysis" section 4.6 to reflect their figure presentation. That is, they should state that the "threshold for statistical significance was set at p<0.053". Since the authors state that morphine caused a "significant increase of both b2 and b5 proteolytic activity at 24 (b2: p=0.053 vs control" (Lines 117-118), then it means that 0.053 is their threshold. However, what is confusing is that a few lines later (Lines 121-123), the authors state that tapentadol did not significantly change the enzymatic activity....(p>0.05 vs control)". So which is it: 0.05 is significant or is 0.053 significant?

Minor:

1) Authors need to have a native English speaking person proof their paper.

Author Response

Reviewer 2

In this manuscript, Rullo et al. examined the effect of opioid agonist exposure on ROS levels, SOD activity/gene expression and proteosome activity in SH-SY5Y cells. The authors found that ROS levels increased after 24 and 48 hr morphine exposure. SOD activity also decreased after 24 hr and not 48. SOD1 gene expression increased significantly at 24 hr, but not 48 hr. Finally, beta2 trypsin-like and b5 chemotrypsin-like activity were increased with morphine but decreased with buprenorphine.

There are some significant issues with this paper. 

1) The authors used a really high morphine and tapentadol concentration. At these levels it is likely that the drugs had nonspecific effects. Further, the time of exposure is rather long because opioid receptor internalization very likely occurred before the 24 hr period expired. 

A   We thank the reviewer for this comment. As reported in the material and methods section, morphine and tapentadol concentrations, as well as the time of cell exposure, were chosen in agreement with previous studies (PMID: 24488603, PMID: 23715695; PMID: 28963507; PMID: 18539596). In addition, we decided to run a new experiment, as showed in the figure 1, in order to evaluate the effects of investigated drugs on ROS production  also at lower exposure times. However, based on results showing ROS alterations at 24 and 48 hours only, we decided to perform the subsequent analysis at these selected time points.

2) To show convincing evidence that the actions were mediated via mu opioid receptors, the authors have to show that they can pharmacologically block the effect of the opioids.

A          We thank the reviewer for his/her comments. As suggested we performed an additional experiment aimed to show the involvement of MOR activation in ROS generation. Since an increase of ROS levels were observed the in morphine-treated cells only, we decided to assess ROS parameter in this specific cell culture at 24 hours according with our previous results.  For this purpose, cells were co-treated with morphine and 10 μM of the non-selective opioids antagonist, naloxone, or with morphine10 μM and 10 μM of the selective MOR antagonist, β-funaltrexamine. As showed in the results (paragraph 2.1; pag.2, lines 91-100) morphine-induced reactive oxygen species generation is completely blocked by the nonselective MOR antagonist naloxone and the selective MOR antagonist β-FNA, thus confirming the involvement of MOR activation in the appearance of this intracellular phenomenon. However, the capacity of opioid antagonists to prevent or revert ROS generation is not surprising since numerous evidence have already highlighted the neuroprotective and scavenger action of naloxone (PMID: 16987355; PMID: 10784126). Even though the opioid antagonist ability to prevent morphine-induced proteasome activation has not been assessed,,  available evidence  shows that naloxone prevents  the increase of proteasome activity in cells treated with 10 μM morphine for 48 hours, thus indicating that chronic morphine exposure enhances proteasome activity through opioid receptor activation.(PMID: 18539596).

The cell type employed has other opioid receptors. How do the authors rule out their involvement when using such high opioid concentrations?

A         We thank the Reviewer for his/her comments. However, in the light of the results obtained in the experiments with antagonists we could be quite sure that the observed effects are MOR-mediated. In particular the use of the selective antagonist β-FNA clearly showed that the ROS production are mainly related to the MOR activation in our experimental conditions.

4) Authors must show scatter plots for all their data. Bar graphs are not convincing.

A          We thank the reviewer for the comments and accordingly we change the graphs in the Results section.

5) The authors should also change their "statistical analysis" section 4.6 to reflect their figure presentation. That is, they should state that the "threshold for statistical significance was set at p<0.053". Since the authors state that morphine caused a "significant increase of both b2 and b5 proteolytic activity at 24 (b2: p=0.053 vs control" (Lines 117-118), then it means that 0.053 is their threshold. However, what is confusing is that a few lines later (Lines 121-123), the authors state that tapentadol did not significantly change the enzymatic activity...(p>0.05 vs control)". So which is it: 0.05 is significant or is 0.053 significant?

A          We thank the Reviewer for the comments and we apologized for the mistake.  As reported in the in “4.6 statistical analysis section” the threshold for statistical significance was set at p < 0.05. In agreement with Reviewer’s comments, we modified the statement and the graph in the results section.

Minor:

1) Authors need to have a native English speaking person proof their paper.

A          We thank the Reviewer and we agree that there were some mistakes. We had a native English speaker who critically read the manuscript and amended many mistakes. Again, thank you for your help.

Round 2

Reviewer 1 Report

The manuscript has been edited following the suggestions of the reviewer. 

The authors have decided to do not undertake additional assays as originally suggested but the manuscript has been rendered less speculative. 

Reviewer 2 Report

The reviewers have addressed my concerns.